# Functionalization of Polylactide with Multiple Tetraphenyethane Inifer Groups to Form PLA Block Copolymers with Vinyl Monomers

**DOI:** 10.3390/ijms24010019

**Published:** 2022-12-20

**Authors:** Mateusz Grabowski, Bartłomiej Kost, Agnieszka Bodzioch, Melania Bednarek

**Affiliations:** Centre of Molecular and Macromolecular Studies, Polish Academy of Sciences, Sienkiewicza 112, 90-363 Lodz, Poland

**Keywords:** polylactide, block copolymer, tetraphenylethane, inifer, radical polymerization

## Abstract

In the present contribution, a new strategy for preparing block copolymers of polylactide (PLA), a bio-derived polymer of increasing importance, is described. The method should lead to multiblock copolymers of lactide with vinyl monomers (VM), i.e., monomers that polymerize according to different mechanisms, and is based on the introduction of multiple “inifer” (*INItiator/transFER agent*) groups into PLA’s structure. As an “inifer” group, tetraphenylethane (TPE, known to easily thermally dissociate to radicals) was incorporated into PLA chains using diisocyanate. PLA that contained TPE groups (PLA-PU) was characterized, and its ability to form initiating radicals was demonstrated by ESR measurements. PLA-PU was used as a “macroinifer” for the polymerization of acrylonitrile and styrene upon moderate heating (85 °C) of the PLA-PU in the presence of monomers. The formation of block copolymers PLA/PVM was confirmed by ^1^H NMR, DOSY NMR, and FTIR spectroscopies and the SEC method. The prepared copolymers showed only one glass transition in DSC curves with T_g_ values higher than those of PLA-PU.

## 1. Introduction

Polylactide is one of the polymers under development that, originating from renewable resources, could replace petroleum-based polymers in some applications [1,2]. Concerning the various sophisticated applications, e.g., in biomedicine, but also in the production of commodity products, the tuning of material properties to meet application requirements is needed. One of the methods of modifying polymer properties is the introduction of other structural units into the polymer macromolecule. This can be achieved in different ways, one of which is copolymerization. Block copolymers are a combination of blocks of two or several homopolymers of different natures that provide new properties in comparison with individual homopolymers, and very often they can self-organize into interesting and sometimes very useful structures. Block copolymers can be synthesized via different approaches. The simplest method is successive polymerization if both blocks can by polymerized by the same mechanism. If, however, two blocks are to be prepared by different mechanisms, such as the copolymer polylactide/poly(vinyl monomer), then ring-opening polymerization has to be combined with known methods of radical polymerization including appropriate block(s) functionalization [3,4,5]. Thus, new combinations of known methods are being searched for and applied.

We have recently described the preparation of the triblock copolymer PLA/poly(acrylonitrile)/PLA [6] using the “macroinifer” (MI) concept [7]. The main step of this synthesis is the preparation of a PLA homopolymer containing a thermal “inifer” group in the middle of the PLA chain [6]. Tetraphenyletahane (TPE) was applied as the inifer, which is known as a group that undergoes dissociation into radicals under heating and as being able to initiate radical polymerization [8,9,10,11]. TPE was generated inside the PLA chain from benzophenone fragments [12,13] via a method we elaborated [6] (commercial benzopinacole could not be introduced via the initiation of lactide polymerization due to the insufficient reactivity of tertiary -OH groups, which was proven by us previously [13]). Using a PLA-TPE macroinifer, acrylonitrile (AN) was polymerized at 85 °C, resulting in the triblock copolymer. As was demonstrated, by varying the molecular weight of the initiating PLA-TPE and the initial AN/PLA-TPE ratio, the composition of the triblock copolymer could be regulated [6].

Following our earlier work, we decided to prepare multiblock copolymers of polylactide and selected poly(vinyl monomers) (PVM) while using the same TPE group as an inifer group but with the introduction of multiple TPE groups into a PLA-based polymer. This was achieved by coupling PLA-diol with TPE-containing diol using aliphatic diisocyanate. A similar approach has been used by other groups for the preparation of multiblock copolymers of poly(tetrahydrofuran), poly(propylene oxide), and poly(ε-caprolactone) with vinyl monomer blocks [14,15,16,17,18,19]. However, the authors used commercial diol containing TPE groups, i.e., tetraphenylethane diol (TPED, benzopinacol), for the coupling reaction. We found that the reaction of such a diol (TPED) with diisocyanate was doubtful due to the insufficient reactivity of the tertiary hydroxyl groups [20]. We also publish the results of experiments aiming to prove this fact, and we suggested replacing the commercial TPED with a derivative synthesized by us. This derivative was synthesized by coupling benzophenone functionalized with primary hydroxyl group [13].

Thus, in the present work, a TPE derivative containing primary hydroxyl groups (TPE-diET, the structure is shown in Figure 1) obtained from 4-hydroxyethoxybenzophenone was coupled with a PLA-diol using diisocyanate to obtain a PLA-based macroinifer. This macroinifer (PLA-PU) was used to polymerize radically acrylonitrile and styrene (ST) to produce multiblock copolymers via the insertion of vinyl monomer units in between TPE units in the PLA chain. The conditions required to achieve control over the synthesis of a polylactide with TPE groups and the course of the radical polymerization are elaborated, and the limitations for the molecular weight of the obtained PLA copolymer are discussed.

## 2. Results and Discussion

### 2.1. Synthesis of Polylactide-Based Polymer Containing Multiple Tetraphenylethane Groups

A PLA-based polymer (polyester-urethane, PLA-PU) was synthesized by coupling PLA-diol with a low molecular weight diol containing TPE (TPE-diET) using hexamethyldiisocyanante (HDI). First, polylactide diol was prepared via the cationic ring-opening polymerization of L-lactide using a difunctional initiator [21]. TPE-diET diol bearing primary -OH groups was synthesized according to the procedure described in our earlier article [13]. As a result of the preliminary experiments, DMF was chosen as a solvent that was able to solubilize all the substrates and products of polyurethane synthesis, and DBTDL was used as a catalyst for the formation of urethane linkages. It was important that the prepared PLA-based polymer had the assumed structure with TPE units covalently linked to the PLA chain. Thus, the synthesis was performed in two steps—first, PLA-diol was reacted with a two-fold excess of HDI in the presence of a catalyst at 40 °C, and next, after about 1.5 h, the TPE-diET was added to a small amount of solvent. The reaction was continued up to 24 h, and the product was analyzed at each reaction step by ^1^H NMR of the reaction mixture. Finally, the product was precipitated two times before final analyses and further use. The success of coupling and the incorporation of TPE units in the polymer structure was confirmed by ^1^H NMR and FTIR analyses. Corresponding spectra are shown in Figure 1 and Appendix A.

In the ^1^H NMR spectra, signals corresponding to all expected units from polylactide, TPE-diET, and HDI are present. Additionally the signal of the formed -NH- groups from urethane at 7.43 ppm appears after the addition of HDI and at 5.72 ppm after the addition of TPE-diET. Signals corresponding to -CHOH PLA end groups and to -CH_2_CH_2_OH groups in TPE-diET were shifted to new positions with higher chemical shift values after the reaction of -OH groups with diisocyanate (Figure 1).

In the IR spectra, the signals corresponding to -NH- groups at 3330 and 3400 cm^−1^, to C=O groups from urethane at ~1730 cm^−1^, and to -C-NH- at ~1630 cm^−1^ [22] are visible in addition to signals characteristic of PLA and TPE-diET (Appendix A). The obtained PLA-based polyurethane (PLA-PU) analyzed by SEC chromatography had a moderate molecular weight (its GPC curve is shown in Appendix A).

### 2.2. The Ability of TPE Groups inside PLA-Based Polyurethane to Form Radicals

Although, according to the above-cited literature, the -C(Ph)_2_-C(Ph)_2_- bond in a benzopinacole molecule can dissociate and initiate radical polymerization, it was not obvious that it could form radicals when present in a benzopinacole moiety (TPE-diET) placed in polylactide-based polyurethane. To check this, electron spin resonance (ESR) spectroscopy measurements were performed. PLA-PU with TPE inside was analyzed in the presence of TEMPO (2,2,6,6-tetramethylpiperidine-1-oxyl) radical able to capture -C(Ph)_2_• radicals formed during heating. Figure 2 presents the ESR spectra recorded at different temperatures and the progress of the decomposition of TPE groups present in PLA-PU polymer with the temperature increase. The higher the temperature, the faster the decomposition of the TPE units (more formed radicals, and, consequently, more trapped radicals). It can be seen that above 75 °C, the rate of dissociation increases only slightly, and thus a temperature of 85 °C was selected in order to generate radicals at a sufficient rate while also limiting possible side reactions.

### 2.3. Polymerization of Acrylonitrile and Styrene Initiated with PLA-PU Containing TPE Groups

The anticipated formation of multiblock polylactide/poly(vinyl monomer) copolymers by the adaptation of the inifer concept [7] is presented schematically in Figure 3. In brief, assuming that, after the initiation of the radical polymerization of VM and several propagation steps, growing radicals will recombine, the introduction of VM blocks into PLA-based PU should proceed by the insertion of VM in the middle of the TPE groups present in PLA-PU.

The idea presented in Figure 3 can be realized under one condition, i.e., when growing radicals are able to recombine. As is known from the literature, the termination of the polymerization of (meth)acrylates proceeds mainly by disproportionation [23,24], so in this case, there is a limited possibility to recombine the growing …PLA-…- C(Ph)_2_-VM• fragments that are formed after the dissociation of the TPE groups and the attachment of several VM units once again into one chain. The opposite is the case in the polymerization of styrene, where the recombination of styryl radicals dominates over their disproportionation [23]. Another vinyl monomer–acrylonitrile has been found to be capable of polymerizing with termination, mainly by recombination under appropriate conditions [25]. Considering this, in the present study, two vinyl monomers were chosen for the polymerization initiated by the PLA-PU macroinifer, i.e., acrylonitrile and styrene. Polymerizations were performed in two solvents with different polarities, DMF and anisole (the normalized solvent polarity parameter E*_T_^N^* = 0.386 for DMF and 0.198 for anisole [26]), by heating PLA-TPE macroinifer in the presence of a vinyl monomer to 85 °C. The dependence of the amount of VM incorporated in the copolymer on the initial VM/PLA ratio was shown in our previous work [6], and thus in this study, polymerizations were performed using the same mass of AN or ST with respect to the PLA-based macroinifer. The products isolated by the double precipitation of the reaction mixture were analyzed by the ^1^H NMR, FTIR, and SEC methods (Figure 4, Appendix A). In the ^1^H NMR spectra of the copolymers prepared in DMA or anisole, the signals corresponding to AN or ST units were clearly visible. (Figure 4 presents exemplary spectra of the products of VM polymerization). By comparison with the intensities of the appropriate signals, the VM/LA ratio could be calculated.

The applied substrates for the vinyl monomer polymerization and characterization of the obtained products are presented in Table 1.

Relatively high conversion of acrylonitrile was achieved in both solvents, DMF and anisole, during the applied polymerization time. It appeared, however, that the polymerization of styrene was not efficient at 85 °C (relatively low conversion), and thus it was also performed at 95 °C. As expected, a higher temperature promoted the decomposition of the TPE groups and, consequently, the conversion of styrene.

The SEC analysis of the radical polymerization products provides two types of information. The first important observation is that all curves show a rather broad but definitely monomodal distribution, indicating the formation of copolymers (both monomeric units corresponding to lactide and vinyl monomer were detected by ^1^H NMR spectroscopy). However, unexpectedly, when polystyrene calibration was applied (no reliable calibration for PLA/PVM copolymers is available), the obtained PLA/PVM copolymers had lower molecular weights than the initial PLA-PU (Table 1). Although the desired result is an increase in the molecular weight, a M_n_ value decrease is also explainable. First, different structural polymers were analyzed by SEC chromatography using unsuitable standards (polystyrene). Another possible reason for the observed decrease in the molecular weight is the sensitivity of polylactide to high temperature, and its possible degradation in the presence of small traces of catalyst and free -OH groups [28,29,30]. An additional experiment was performed where polyurethane prepared from PLA alone and HDI was heated for 24 h in DMF. It appeared that the molecular weight of PU dropped from 36,000 to 7000 as determined by SEC. In the ^1^H NMR spectrum, new signals appeared corresponding to newly formed -OH groups (-CHOH and -OH). The results are presented in the Appendix A. It is not excluded that DMF also contributed somehow to the PLA decomposition, because when the same PU prepared from PLA diol was heated in anisole, decomposition products were not detected in the ^1^H NMR spectrum (although the M_n_ determined by SEC also decreased).

It should also be considered that a molecular weight decrease is a result of the incomplete recombination of growing -VM• radicals. If some growing radicals terminate by recombination and others by disproportionation, the products would be a mixture of multiblock copolymers of different lengths, which could explain the relatively broad dispersity observed by SEC. Because PLA-PU used as an inifer contains many TPE units, if the contribution of termination mechanism other than the recombination is not insignificant, the molecular weight of the final product may indeed be lower than that of the initial molecular weight of the PLA-PU.

Because the results obtained from the SEC analysis could not be treated as fully reliable, we searched for another method by which we could confirm the formation of copolymers. Such a possibility is offered by the application of diffusion-ordered spectroscopy (DOSY) NMR [31]. By this NMR method, each component in a mixture can be virtually separated based on its own diffusion coefficient. DOSY NMR appeared to be very useful to study the polymerization kinetics and determine the structure and molecular weight of the formed product [32,33,34]. Figure 5 presents DOSY maps (with ^1^H NMR spectra on the top) obtained for the PLA-PU/PAN copolymer, the PLA-PU starting polymer, and the separately prepared PAN homopolymer. Similarly, in Figure 6, DOSY maps for the PLA-PU/PST copolymer, the PLA-PU, and the PST homopolymer are shown.

As can be seen in Figure 5 and Figure 6 (and the data shown in Appendix A), the diffusion coefficients for building blocks in pairs—PLA-PU, PAN (D_av._ = 1.26·10^−7^ cm^2^/s and 0.78·10^−7^ cm^2^/s, respectively) and PLA-PU, PST (D_av._ = 1.43·10^−7^ cm^2^/s and 7.47·10^−7^ cm^2^/s, respectively)—differ. Additionally, the values of the diffusion coefficients of the radical polymerization products are not the same as those characteristic of their building blocks (i.e., D_av_ = 2.81·10^−7^ cm^2^/s for PLA-PU/PAN and 3.22·10^−7^ cm^2^/s for PLA-PU/PST), confirming the formation of the copolymer. It should be noted from the DOSY NMR spectra that the same diffusion coefficient for different structural components of PLA-PU means that the used macroinifer was pure (i.e., not contaminated with PLA diol or TPE-diET diol).

An analysis of the products by DOSY NMR provides unequivocal proof that they are copolymers because signals of both PLA-PU and PVM blocks lie on the same diffusion line, showing that they are fragments of the same macromolecule. Moreover, in the spectra of the copolymers, there are no signals for which the diffusion coefficient would correspond to the diffusion coefficients of the starting PLA-PU or ST or AN homopolymers, which indicates the efficient initiation of VM polymerization by PLA-PU inifer and that the products are not contaminated by corresponding homopolymers.

The prepared PLA/PVM copolymers were also analyzed using the DSC method (Figure 7).

In the DSC curves of the copolymers, only one glass transition is visible, which means that there are no separated phases from the building blocks [35]. In the synthesized copolymers, both blocks are relatively short, and thus phase separation does not occur. On the other hand, the T_g_ values of the PLA/PAN and PLA/PST copolymers are different. They are higher than the T_g_ of the PLA-PU macroinifer. As the vinyl monomer fraction increases (polymerization of styrene performed at a higher temperature), the T_g_ value increases because the T_g_s of the homopolymers of PAN and PST are higher (~96 °C [36] and ~100 °C [37], respectively, for polymers with an infinitive molecular weight). The DSC results also confirm the formation of the PLA/PVM copolymer.

## 3. Materials and Methods

### 3.1. Materials

L,L-Lactide from Purac was recrystallized from 2-propanol, sublimated, and stored under vacuum. The 2-bromoethanol (95%), 4-hydroxybenzophenone (HBP, 98%), hexamethylene diisocyanate (HDI, 99%), ethylene glycol (EG, 99.8%), which were all from Sigma-Aldrich (St. Luis, MO, USA/Steinheim, Germany), were used as received. Acrylonitrile (AN, 99%) was passed through an inhibitor removal column filled with silica gel and distilled under a vacuum. Styrene was purified with basic Al_2_O_3_ to remove the inhibitor and distilled under a vacuum. Trifluoromethanesulfonic acid (triflic acid, 99%) from Sigma-Aldrich was distilled under vacuum, and dibutyltin dilaurate (DBTDL, 98%) from ABCR was used as received. Dichloroethane (DCE, pure p.a.), dimethylformamide (DMF, pure p.a.), and anisole (pure p.a.) from POCH (Gliwice, Poland) were dried over CaH_2_, distilled before use, and stored over molecular sieves (4 Å). Chloroform (pure p.a.), 2-propanol (pure p.a.), ethanol (pure p.a.), diethyl ether (pure p.a.), and hexane (99%), all from POCH (Gliwice, Poland), were used as received. Potassium hydroxide (pure p.a.) was obtained from POCH (Gliwice, Poland), magnesium sulfate (MgSO_4_, 99%) was obtained from Chempur (Piekary Slaskie, Poland), and both were used as received.

### 3.2. Synthesis of Polylactide Diol

Four grams (20 mmol) of L-lactide together with a stirring bar was placed in a Schlenk flask, which was evacuated for 0.5 h and then backfilled with N_2_. Then, 9 mL of DCE was introduced through the rubber septum with a syringe, followed by the addition of ethylene glycol as an initiator (220 μL, 3.89 mmol) and triflic acid as a catalyst (100 μL, 1.13 mmol). Polymerization was performed at room temperature for 24 h. Then, the reaction mixture was neutralized by the addition of CaO, was filtered from the CaO, and was precipitated into hexane. 

### 3.3. Synthesis of TPED Derivative Containing Primary Hydroxyl Groups (TPE-diET)

A total of 3.36 g (86 mmol) of potassium hydroxide was dissolved in 5 mL of distilled water in a beaker. Then, the solution was added to 10 g (50 mmol) of 4-hydroxybenzophenone in a round bottom flask. The resulting yellow reaction mixture was stirred at 70 °C until the HBP was dissolved. Next, 8 mL (108 mmol) of 2-bromoethanol was added via a syringe, and the stirring was continued for 24 h. The aqueous phase was subjected to extraction with chloroform, and the organic phase was extracted with distilled water 3 times. The combined organic extracts were dried with MgSO_4_. The solvent was evaporated under a vacuum, and the resulting product was recrystallized by dissolving it in 4 mL of ethanol while heating. Then, excess diethyl ether was added and cooled down, which resulted in crystal growth. White crystals of HBP-ET were dried under vacuum and obtained with a yield of ~50%. In the next step, HBP-ET was placed into a quartz flask to which isopropanol (8 mL for every 1 g) was added. The flask was closed with a rubber septum and purged with nitrogen (with a needle) for 15 min. Then, the flask content was slightly heated until the mixture was dissolved. The homogenous solution was exposed to ultraviolet light (365 nm) for 24 h. The resulting white precipitate TPE-diET was then centrifuged from the rest of the reaction mixture and dried under vacuum (>99% pure, ^1^H NMR, Figure 1).

### 3.4. Synthesis of PLA-Based Polyurethane (PLA-PU) Containing Multiple TPE-diET Groups

One gram of PLA-diol (1.18 mmol) was placed in a Schlenk flask that was evacuated for 0.5 h and then backfilled with N_2_. Next, 5 mL of DMF was added and the mixture was stirred until the polymer dissolved. A total of 0.40 mL (2.48 mmol) of HDI was introduced through the rubber septum with a syringe followed by the addition of DBTDL (36 μL, 0.94 mmol) as a catalyst, and the flask was immersed in an oil bath preheated to 40 °C. After 1.5 h, when the conversion of PLA end groups was complete according to ^1^H NMR, 0.46 g of TPE-diET dissolved in 1.5 mL of DMF was added through the rubber septum with a syringe. The mixture was stirred for 24 h total. Next, the reaction mixture was precipitated twice to cool diethyl ether. The resulting polymer was dried under a vacuum. The product was obtained with a yield of ~90%.

### 3.5. Polymerization of Acrylonitrile or Styrene Using PLA-PU as a Macroinifer

The synthesis of the PLA/PAN or PLA/PST copolymer was performed as follows: PLA-PU (0.3 g) was placed in a Schlenk flask and vacuumed, and the flask was filled with nitrogen. Next, 3 mL of DMF (or anisole) was added through the rubber septum, the mixture was stirred until the polymer dissolved, and subsequently 0.32 mL of AN (0.3 g, 4.7 mmol) or 0.33 mL of ST (0.3 g, 2.9 mmol) was added to the reaction flask. The solution was degassed via three freeze–pump–thaw cycles. The flask was filled with nitrogen and heated in an oil bath at 85 °C for 24 h with stirring. The resulting polymer was precipitated twice to cold diethyl ether, decanted, and dried under reduced pressure.

### 3.6. Instrumental Methods

The ^1^H NMR spectra of the synthetized polymers were recorded in DMSO-d_6_ (or CDCl_3_) using a Bruker Avance 400 Neo instrument (Bruker, Billerica, MA, USA) operating at 400 MHz. 

DOSY experiments were performed in DMSO-d_6_ on a Bruker DRX500 spectrometer operating at 500 MHz (11.7 T).

Size exclusion chromatography (SEC) was performed in DMF as a solvent at 40 °C using a Wyatt (Dernbach, Germany) instrument equipped with two perfect separation solution (PSS) columns and one guard column (GRAM Linear, 10 μm, M_n_ between 800 and 1,000,000 Da) with a differential refractometer detector. The measurements were performed at a flow rate of 1 mL/min of the DMF eluent, containing 50 mmol LiBr (calibration with polystyrene standards). Alternatively, some analyses were performed using a Shimadzu Pump LC-20AD with a Shimadzu DGU-20A5 Degasser (Kioto, Japan) with a DMF flow rate of 0.8 mL·min^−1^ (polystyrene standards). 

Electron spin resonance (ESR) spectra were recorded on an X-band EMX-Nano ESR spectrometer equipped with a nitrogen variable temperature unit on degassed DMF. The microwave power was set with the Power Sweep program below the saturation of the signal: power attenuation of 20 dB, modulation frequency of 100 kHz, modulation amplitude of 0.5 Gpp, and spectral width of 200 G. ESR spectra were recorded every 10 °C during the heating of the measuring tube over the range 35–85 °C with a tolerance of 2 °C. After reaching each measurement temperature, the system was stabilized for 30 s, during which time the spectrum was recorded.

Fourier transform infrared spectroscopy (FTIR) measurements were performed on a Thermo Scientific Nicolet 6700 instrument with an attenuated total reflectance (ATR) GoldenGate accessory (Waltham, MA, USA) and deuterated triglycine sulfate (DTGS) or mercury–cadmium–telluride (MCT) detectors. The spectra were obtained by adding 64 scans at a 2 cm^−1^ resolution. 

Differential scanning calorimetry (DSC) analysis was performed at temperatures from 0 °C to 200 °C on a DSC 2500 TA Discovery system. The analyses were carried out in a nitrogen stream at heating and cooling rates of 10 °C min^−1^.

UV coupling was performed in a photochemical reactor (RayonetRPR-200, Southern New England, Brandford, CT, USA) equipped with 12 UV lamps (λ = 365 nm) and a magnetic stirrer.

## 4. Conclusions

A polylactide-based macroinifer was successfully prepared via the introduction of a thermally dissociating tetraphenylethane group into a PLA chain byw coupling PLA diol with TPE-containing diol using diisocyanate. The coupling reaction was possible because the TPE derivative we synthesized contained primary hydroxyl groups. The coupling reaction appeared to be the most efficient for the two-step process performed with DMF as a solvent and DBTDL as a catalyst. The ability to form radicals upon heating of PLA-PU macroinifer containing multiple TPE groups was confirmed by ESR measurements. PLA-based macroinifer was applied to initiate the radical polymerization of acrylonitrile and styrene under heating at 85–95 °C. Although higher temperatures promote the TPE groups’ decomposition to initiating radicals, it was found that 85 °C was sufficient to achieve up to 85% conversion of the monomer (used in the same mass amount as PLA-PU) over 24 h. The presence of new monomer units in the formed polymeric product was confirmed by ^1^H NMR and FTIR spectroscopy. Moreover, the formation of the copolymer was unequivocally proven via DOSY NMR experiments. In the SEC curves, a decrease in the molecular weight could be assigned to the inadequate conditions of SEC analysis, the thermal instability of PLA chains, and the termination of growing radicals proceeding not only by recombination but partially by disproportionation. Despite the limitation on the value of the molecular weight of the obtained PLA/PVM copolymers, we proved that the described, relatively simple, method based on TPE inifer leads to copolymer formation. The study will be continued to elaborate conditions for the preparation of copolymers with higher molecular weights and higher block lengths, which could result in thermal property modification (e.g., an increase in the T_g_ value) or interesting morphologies (e.g., phase separation useful for obtaining porous materials).

## Figures and Tables

**Figure 1 ijms-24-00019-f001:**
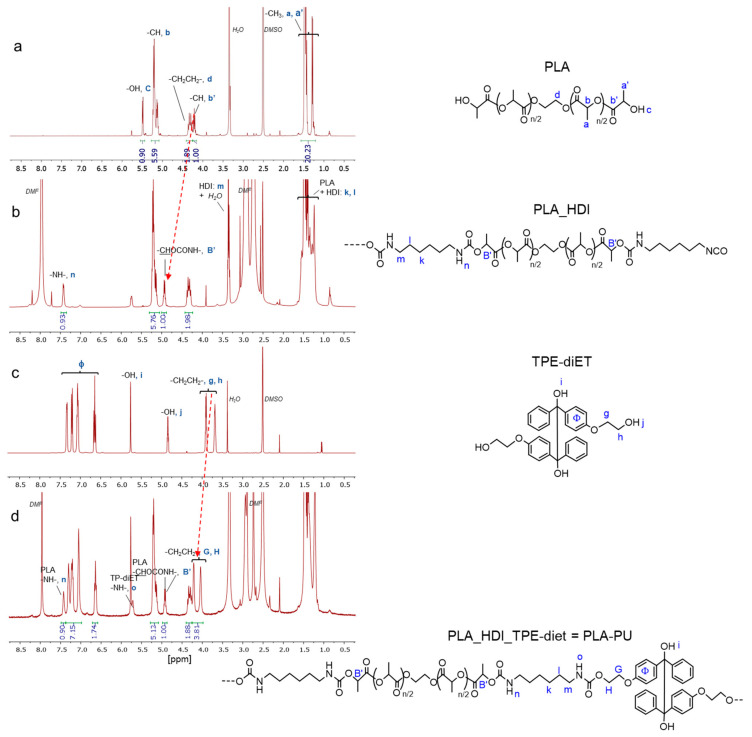
Tracking the formation of PLA-based polyurethane (PLA-PU) containing TPE groups by ^1^H NMR analysis (DMSO-d_6_). Red arrows indicate changes in the position of the terminal -CHOH (from PLA: (**a**,**b**)) or -CH_2_CH_2_OH (from TPE-diET: (**c**,**d**)).

**Figure 2 ijms-24-00019-f002:**
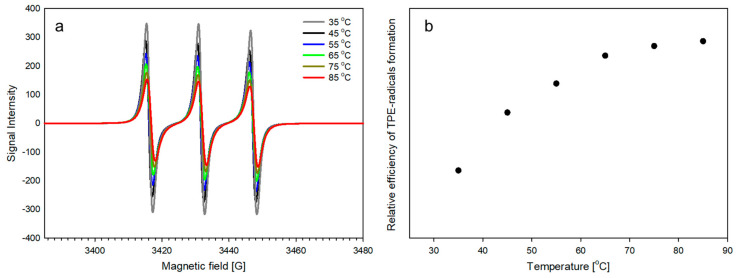
(**a**) ESR spectra of PLA-PU recorded at different temperatures in the presence of TEMPO. (**b**) Progress of the decomposition of TPE groups in a PLA-PU polymer with an increase in temperature (the relative concentration of radicals was calculated according to the method described in the Appendix A using normalized intensities from ESR spectra, Appendix A).

**Figure 3 ijms-24-00019-f003:**
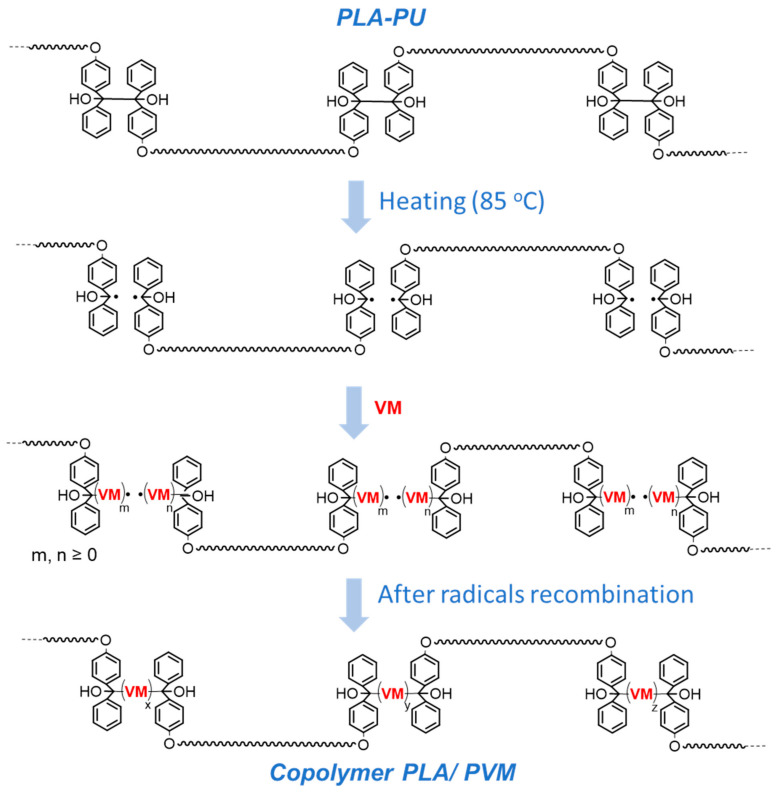
Schematic representation of the formation of a PLA/PVM copolymer using a PLA-PU macroinifer.

**Figure 4 ijms-24-00019-f004:**
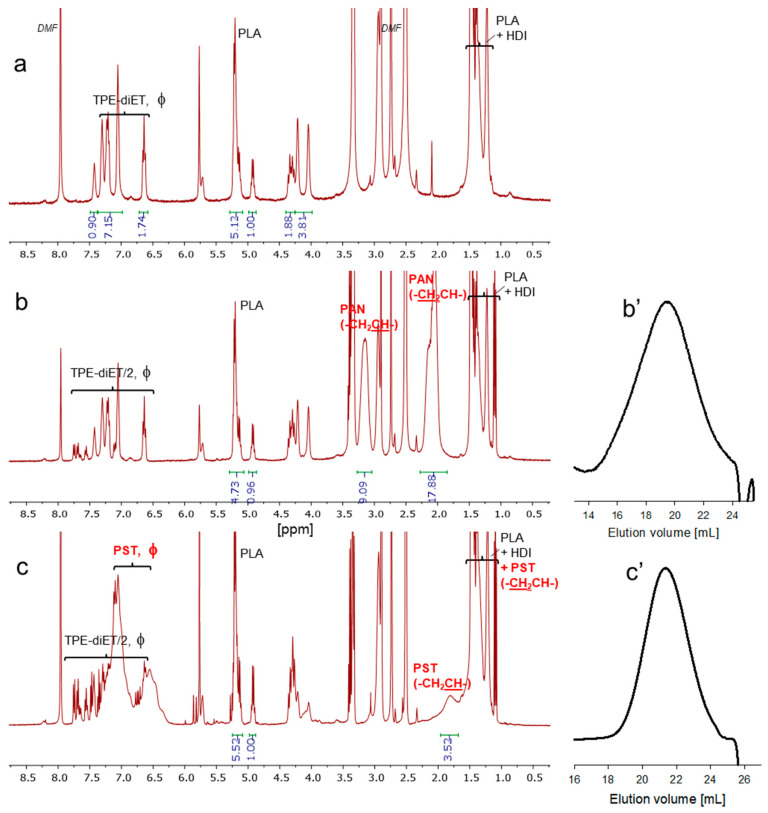
^1^H NMR spectra of the PLA-PU (**a**), PLA/PST (**b**), and PLA/PAN (**c**) copolymers prepared in DMF or anisole as a solvent; (**b’**,**c’**)—corresponding SEC curves.

**Figure 5 ijms-24-00019-f005:**
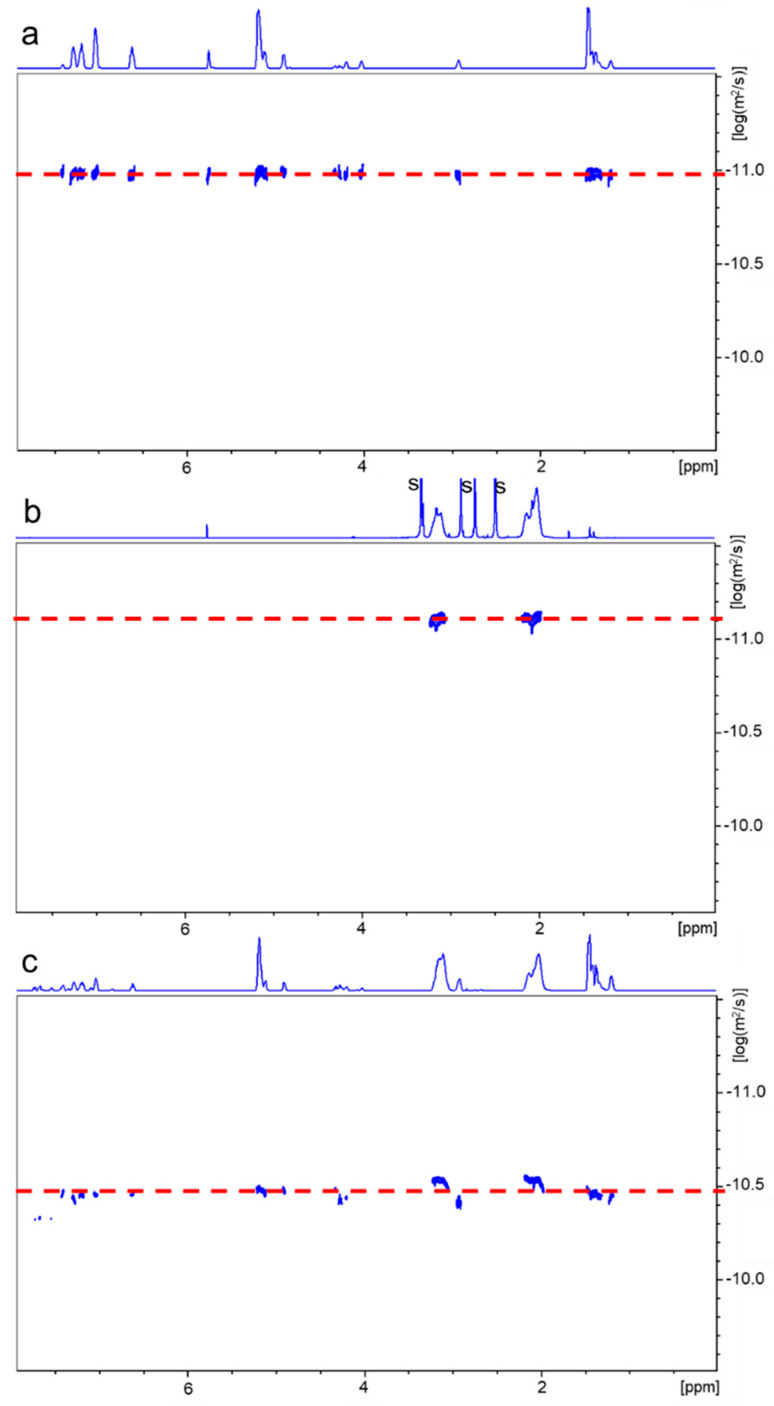
2D DOSY NMR spectra of PLA-PU (**a**), PAN (**b**), and PLA/PAN copolymer (**c**) recorded in DMSO-d_6_ (the area of the appearance of signals corresponding to the remaining solvents “s” is not shown).

**Figure 6 ijms-24-00019-f006:**
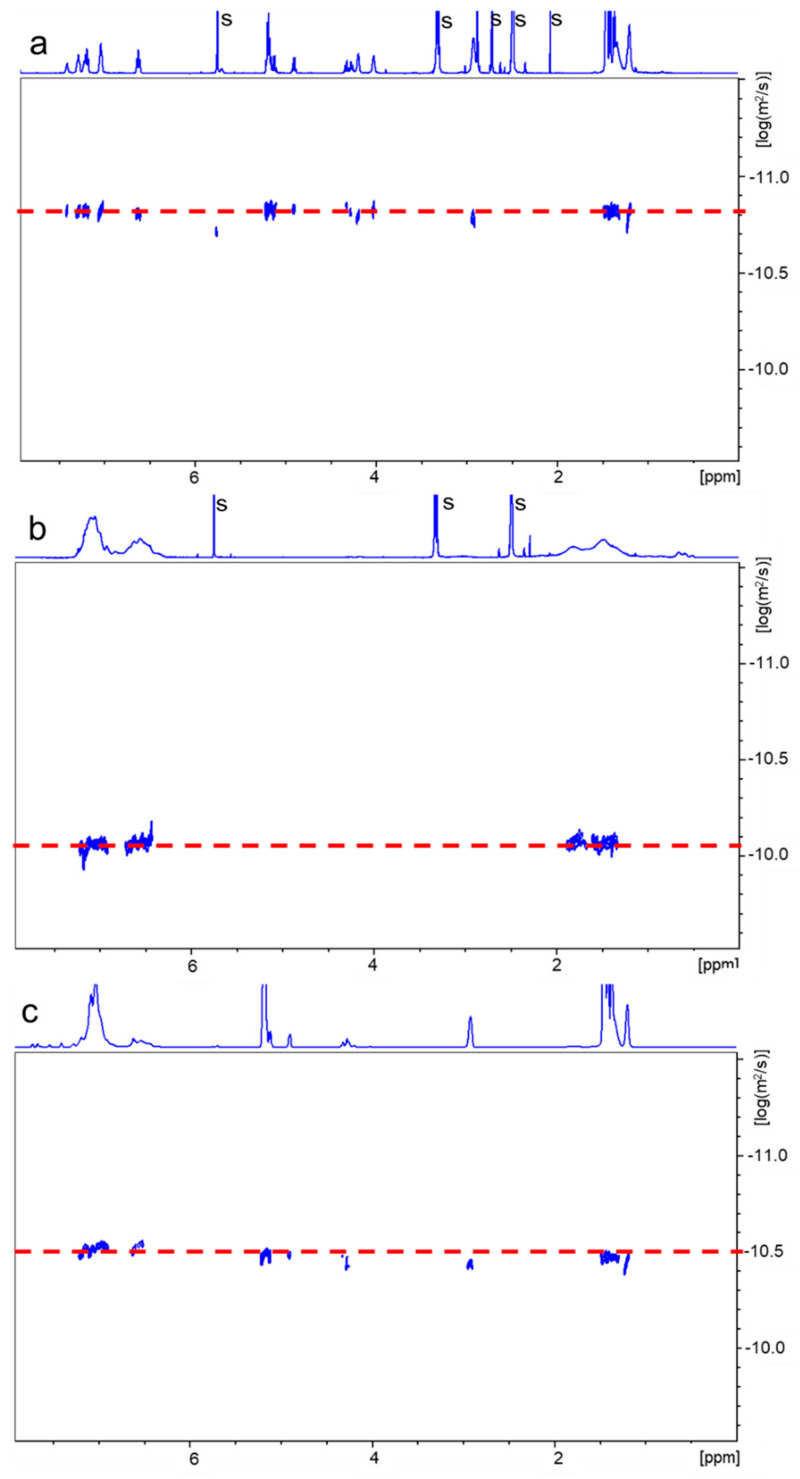
2D DOSY NMR spectra of PLA-PU (**a**), PST (**b**), and PLA/PST copolymer (**c**) recorded in DMSO-d_6_ (the area of the appearance of signals corresponding to the remaining solvents “s” is not shown).

**Figure 7 ijms-24-00019-f007:**
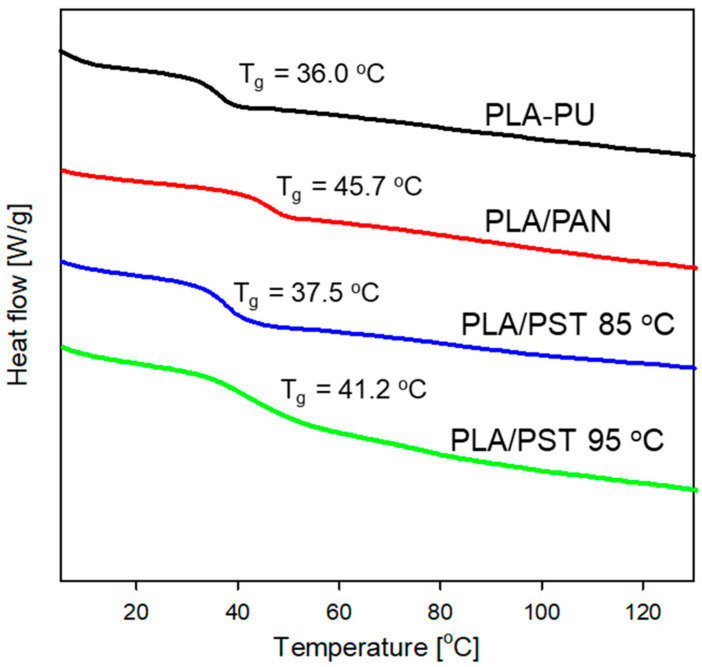
DSC curves for the PLA-PU macroinifer and the synthesized PLA/PAN copolymers: PLA/PAN, PLA/PST85, and PLA/PST95 (positions 2, 6, and 4 in Table 1, respectively). Fragments of DSC thermograms are shown where glass transitions could appear.

**Table 1 ijms-24-00019-t001:** Polymerization of VM with PLA-PU macroinifer (M_n_ = 37000, SEC), 85 °C, 24 h; the mass ratio of VM/PLA in the feed = 1.

Substrates				Conversion of VM [%]	PLA/PVM Copolymers ^(a)^	
Vinyl Monomer, VM	Polymeriz. No	Mol Ratio of VM/LA in Feed	Solvent		Mol Ratio of VM/LA, ^1^H NMR	M_n_, SEC ^(b)^
AN	1	4.8	DMF	85	7.0	n.d.
	2	4.8	Anisole	70	3.8	16,600
ST	3	2.5	DMF	37	0.4	n.d.
	4	2.5	DMF	45 (95 °C)	1.2	13,000
	6	2.5	Anisole	33	0.5	11,600

^(a)^ Products analyzed after double precipitation (50–60% yield). ^(b)^ Calibration for polystyrene; M_n_ values should be treated with caution because polystyrene calibration could be inappropriate for the calculation of the M_n_ of a copolymer of LA with VM. The molecular weights determined for PLA with PST standards differ (sometimes significantly) from real values, thus different correcting factors are often applied [27]. n.d.—not determined.

## Data Availability

Not applicable.

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
