# Peer review of "Functionalization of Polylactide with Multiple Tetraphenyethane Inifer Groups to Form PLA Block Copolymers with Vinyl Monomers"

_ijms, 2022, doi:10.3390/ijms24010019_

Round 1

Reviewer 1 Report

The manuscript written by prof. Melania Bednarek et al. presents a strategy for synthesis of block copolymers composed of blocks that can be prepared by different mechanism, i.e. block copolymers PLA/poly(vinyl monomer) (vinyl monomer = acrylonitrile and styrene). The authors proposed synthesis of polylactide diol by cationic ring opening polymerization, followed by preparation of  PLA-based macroinifer containing TPE (tetraphenylethane) units able to form radicals and initiate radical polymerization of vinyl monomers. The synthesized compounds were characterized by spectroscopic  (1H NMR and FT-IR) and SEC methods, and the structure of block copolymers was also confirmed by 2D DOSY NMR and DSC analysis. The experimental results are interesting and there is an attempt for their interpretation and explanation. The presentation of the results is logical and clear. Authors should however, carefully check and correct the References part (for many items the name of the journal and page numbers are missing). The manuscript can be accepted for publication.

Author Response

Thank you for your comments.

The answer to the comment “Authors should however, carefully check and correct the References part (for many items the name of the journal and page numbers are missing).”

References are corrected.

Reviewer 2 Report

Dear Researcher,

PLA is one of the mostly studied polymer nowdays.This is excellent work touched all aspect of polymer science and engineering. In conclusion can you comments about what is end application of this polymer and futher need to modify for future researcher.

Few comments

1.Did you study dynamic mechanical analysis,XRD (wide or small angle) that will be additional characteristics(based on theme of journal)?

2.Can you add mechanism of polymerization in reference (though it well known in the field),can be useful for future researcher on same topic?

Author Response

Thank you for your comments.

The answer to comments:

  • In conclusion can you comment about what is end application of this polymer and further need to modify for future researcher.
  • Did you study dynamic mechanical analysis,XRD (wide or small angle) that will be additional characteristics(based on theme of journal)?

The work described in the manuscript concerns the first study on the possibility of obtaining block copolymers of polylactide with blocks obtained by a totally different (radical) mechanism by using the not applied so far method (and not applied “inifer” group). To obtain copolymers with well determined structures and for easy analysis, copolymers with relatively low molecular weights and with small block lengths (with a view to confirming the assumed structure) were intentionally synthesized. Such copolymers are not yet usable materials and only lead to them. Thus, DMA and XRD studies were not performed. Since our future research includes the synthesis of copolymers with longer PLA/PVM blocks, we plan to perform such analyses. 

Additionally, In Conclusions, the following sentence is added instead of previous last sentence:

“The study will be continued to elaborate conditions for the preparation of copolymers with higher molecular weights and higher block lengths which could result in thermal properties modification (e.g., increase in the Tg value) or interesting morphologies (e.g., phase separation useful for obtaining porous materials).”

The answer to the comment: “Can you add mechanism of polymerization in reference (though it well known in the field), can be useful for future researcher on same topic?”:

The reference to the “inifer (“macroinifer”) concept (present in the submitted text on page 2 as Ref. 7) is now also placed by the description of operating in our study mechanism on page 9.